

# The geometry of meaning: evaluating sentence embeddings from diverse transformer-based models for natural language inference

Mohammed Alsuhaibani

Department of Computer Science, College of Computer, Qassim University, Buraydah, Saudi Arabia

## ABSTRACT

Natural language inference (NLI) is a fundamental task in natural language processing that focuses on determining the relationship between pairs of sentences. In this article, we present a simple and straightforward approach to evaluate the effectiveness of various transformer-based models such as bidirectional encoder representations from transformers (BERT), Generative Pre-trained Transformer (GPT), robustly optimized BERT approach (RoBERTa), and XLNet in generating sentence embeddings for NLI. We conduct comprehensive experiments with different pooling techniques and evaluate the embeddings using different norms across multiple layers of each model. Our results demonstrate that the choice of pooling strategy, norm, and model layer significantly impacts the performance of NLI, with the best results achieved using max pooling and the L2 norm across specific model layers. On the Stanford Natural Language Inference (SNLI) dataset, the model reached 90% accuracy and 86% F1-score, while on the MedNLI dataset, the highest F1-score recorded was 84%. This article provides insights into how different models and evaluation strategies can be effectively combined to improve the understanding and classification of sentence relationships in NLI tasks.

## INTRODUCTION

Textual entailment (TE), or natural language inference (NLI), is a foundational task in natural language processing (NLP). It involves determining whether a given hypothesis logically follows from a premise. For instance, given the premise "a person is reading a novel" the hypothesis "someone is engrossed in a book" can be inferred. This process requires an understanding of both syntactic structure (*e.g.*, clause dependencies, grammatical roles) and semantic relationships (*e.g.*, inference, paraphrasing, entailment). NLI has various applications in NLP, including question answering (*Mishra et al., 2021*), information retrieval (*Kim et al., 2022*), sentiment analysis (*Song et al., 2020*), machine translation (*Kann et al., 2022*), dialogue systems (*Dziri et al., 2022*) and text summarization (*Laban et al., 2022*).

Corresponding author
Mohammed Alsuhaibani,
m.suhibani@qu.edu.sa

Recent studies have highlighted the importance of sentence embeddings for NLI (*Alsuhaibani, 2023*; *Kowsher et al., 2023*). Sentence embeddings encode sentences into fixed-length vectors, capturing their semantic meanings (*Sato et al., 2024*). This representation allows for the efficient comparison of sentences, crucial for entailment detection. Traditionally, sentence embeddings relied on shallow models like word embeddings aggregated into sentence-level representations. However, these approaches often struggled with complex syntactic structures and semantic relationships. Deep learning models, particularly transformer-based architectures (*Devlin et al., 2018*; *Brown et al., 2020*; *Yang et al., 2019*; *Liu et al., 2019*), have provided more advanced methods for sentence embeddings. These models are capable of capturing linguistic patterns by considering the bidirectional context of words in a sentence.

The utilization of the bidirectional encoder representations from transformers (BERT) sentence embeddings for entailment recognition was recently investigated (*Alsuhaibani, 2023*). Various layers within BERT were explored to identify the best layer for extracting sentence embeddings. Unlike traditional methods, this approach focused on the simple geometric properties of the embeddings, specifically by directly comparing sentence norms to evaluate the NLI. The results demonstrated the effectiveness of this geometric approach, highlighting the potential of sentence embeddings in detecting NLI.

While BERT has shown good results in that study, it covers only one way to enhance NLI. The aspects of NLI suggest that a more comprehensive exploration of various transformer-based models could provide deeper insights. The geometry of sentence embeddings, including different norms and pooling strategies, plays a pivotal role in understanding the nature of entailment. We believe that there is a compelling need for more investigation that not only compares multiple transformer-based models but also systematically evaluates various norms and pooling strategies to uncover the most effective configurations for NLI. Thus, this article aims to conduct a more expansive study, examining diverse transformer models and comparing several norms to refine the evaluation process.

We propose to extend the scope of sentence embedding analysis for NLI. Building on the foundations established by *Alsuhaibani (2023)*, we aim to explore the geometric aspects of sentence embeddings derived from a range of transformer-based models such as BERT (*Devlin et al., 2018*), Generative Pre-trained Transformer (GPT) (*Brown et al., 2020*), robustly optimized BERT approach (RoBERTa) (*Yang et al., 2019*) and XLNet (*Liu et al., 2019*). By comparing multiple norms and pooling strategies, we seek to identify the configurations that best capture the essence of entailment. Our study utilizes the Stanford Natural Language Inference (SNLI) (*Bowman et al., 2015*) dataset. The SNLI dataset is widely recognized for its diverse and well-annotated collection of sentence pairs, each categorized into one of three classes: entailment, contradiction, or neutral. It allows us to ensure a robust evaluation, offering a comprehensive benchmark for assessing the performance of different models and norms.

The main contributions of this article are summarized as follows:

- Extending beyond the existing focus on BERT by evaluating a diverse set of transformer-based models for NLI, providing a broader perspective on each model's strengths and limitations in capturing semantic relationships.
- Exploring various geometric properties of sentence embeddings, including different norms (L1, L2, and L-infinity) and pooling strategies, to identify the most effective configurations for NLI.
- Utilizing a straightforward evaluation approach for all the three NLI scenarios (entailment, contradiction and neutral) by measuring textual entailment by directly comparing the norms of sentence embeddings. This method highlights the geometric attributes of sentence embeddings, providing an efficient alternative to more complex evaluation techniques.
- Analyzing the performance of different layers within transformer models to offer insights into the optimal layer selection for extracting sentence embeddings that best capture NLI.
- Utilizing the extensive SNLI dataset, and a domain-specific dataset (MedNLI), to conduct a thorough evaluation across a wide range of sentence pairs for various levels of textual complexity in NLI tasks.

The remainder of this article is structured as follows: 'Related Work' highlights the related work. 'Proposed Approach' describes our proposed approach, encompassing various models and norms. In 'Experiments and Results', we present our experiments and discuss the results. Finally, the 'Conclusion' concludes the article.

## RELATED WORK

NLI often involves determining how two sentences are related, specifically whether one sentence (the hypothesis) can be inferred, contradicted, or remains neutral in relation to the other (the premise) (*Androutsopoulos & Malakasiotis, 2010*). This task was considered challenging because it required a deep understanding of the meaning of both sentences and how they are connected.

A common approach for addressing NLI tasks is by using sentence embeddings, which are vector-based representations that encapsulate the meaning of sentences (*Yu & Jiang, 2016*). These embeddings can then be employed to train models capable of predicting the relationship between pairs of sentences. There are numerous methods available for generating sentence embeddings. A frequently used technique involved creating word embeddings (*Pennington, Socher & Manning, 2014*; *Alsuhaibani et al., 2018*; *Maillard, Clark & Yogatama, 2019*), and then combine these word-level vectors into a sentence-level embedding. Another popular approach is to use deep learning models specifically trained to produce sentence embeddings (*Pagliardini, Gupta & Jaggi, 2018*; *Arora, Liang & Ma, 2019*; *Gao, Yao & Chen, 2021*; *Jiang et al., 2023*).

Transformer-based models emerged as widely used models for generating sentence embeddings (*Taneja, Vashishtha & Ratnoo, 2023*). They are pre-trained on a large text

*corpus*, enabling them to effectively capture the meaning of words and sentences, making them useful for a variety of NLP tasks, including NLI. When many of these models were first developed, they were trained on tasks like predicting the next word and filling in masked words. This approach enabled the models to learn meaningful representations of both words and sentences. This pre-training has made transformer-based models highly effective for many NLP applications, including NLI.

For example, *Lin & Su (2021)* studied how well BERT handles NLI tasks, particularly how it manages potential biases in the data. In their research, they designed a straightforward entailment task using binary predicates in English. They found that BERT's learning was slower than expected, but incorporating task-specific features enhanced its efficiency. This underscored the value of incorporating domain-specific knowledge for NLI tasks. Prior to this, *Baudis, Stanko & Sedivy (2016)* introduced a joint model that learns sentence embeddings for both relevance and entailment in an information retrieval context. They proposed a system that integrates multiple pieces of evidence to determine whether a hypothesis is true or not. Their approach trains sentence embeddings for both relevance and entailment without explicit per-evidence supervision. This work highlights the potential of integrating retrieval and entailment systems to improve reasoning tasks.

*Gajbhiye, Moubayed & Bradley (2021)* introduced an External Knowledge Enhanced BERT for Natural Language Inference (ExBERT), a model designed to enhance BERT's reasoning capabilities by incorporating external commonsense knowledge. ExBERT enriches the contextual representations produced by BERT by integrating information from external knowledge sources, such as knowledge graphs, to better inform the model's reasoning process. Similarly, *Pang, Lin & Smith (2019)* proposed a technique to integrate syntactic information into NLI models. They used token-level contextual representations generated from a pre-trained dependency parser. This approach, like other contextual embedding methods, is compatible with various neural models, including BERT. Their experiments showed improved accuracy on standard NLI datasets.

Moreover, *Cabezudo et al. (2020)* explored methods to improve inference recognition on the ASSIN dataset (*Fonseca et al., 2016*), which is focused on entailment in Portuguese. Their research included the use of external datasets, such as multilingual datasets or corpora that have been automatically translated, to improve training. Their experiments with a multilingual BERT model showed improvements on the ASSIN dataset, though they found that adding external data did not significantly boost performance. *Wehnert et al. (2022)* introduced three distinct approaches for NLI classification. The first approach combined Sentence-BERT embeddings with a graph neural network. The second approach utilized LEGAL-BERT, a model fine-tuned specifically for entailment classification. The third method employed KERMIT to encode syntactic parse trees, which were then integrated into BERT. Their results indicated that LEGAL-BERT outperformed the graph-based approach, particularly for legal text entailment tasks.

*Shajalal et al. (2022)* introduced a new approach to NLI that uses an empirical threshold-based feature. This feature helps to understand the relationships between a text and its hypothesis. Their experiments on the SICK-RTE dataset (*Marelli et al., 2014*) using

various machine learning algorithms showed that their method improved the model's ability to capture semantic entailment relationships. *Jiang & de Marneffe (2019)* tackled a common problem in NLI datasets by reworking the CommitmentBank (*De Marneffe, Simons & Tonhauser, 2019*) for NLI tasks. Their approach examined the extent to which speakers are committed to the complements of clause-embedding verbs, particularly in contexts that negate entailment. This resulted in hypotheses that were more naturally aligned with the premises and free from dataset-specific artifacts. Although their BERT-based model achieved solid results, they noted that it still struggled with certain linguistic details, especially in pragmatic reasoning.

While many of the approaches mentioned above have advanced the NLI tasks, they often require complex designs. In response, and building on the findings of *Alsuhaibani (2023)*, our work takes a simpler approach by focusing on sentence embeddings and transformer-based models capabilities. We examine the impact of using different layers from transformer-based models like BERT, GPT, RoBERTa, and XLNet to extract sentence embeddings. By evaluating entailment through a straightforward comparison of sentence norms, we focus on the geometric properties of these embeddings. This approach provides a simple and effective approach for NLI tasks. It is grounded in the idea that the norm of a sentence embedding can reflect the cumulative semantic information it encodes. Prior work in geometric and probing-based NLP (*Kobayashi et al., 2020*; *Ethayarajh, 2019*) has shown that vector norms often correlate with sentence complexity or informativeness. Thus, comparing norms offers a lightweight but interpretable proxy for inferring semantic relationships such as entailment.

## PROPOSED APPROACH

We propose a comprehensive and a straightforward approach for NLI by comparing sentence embeddings generated by several transformer-based models, including BERT, GPT, RoBERTa, and XLNet. The embeddings are then processed using various norm-based measures (L1, L2, and L-inf norms) and pooling techniques (max, min, and mean pooling). This approach ensures an evaluation of different models and methods to detect NLI.

We utilize four pre-trained models, BERT, GPT-3, RoBERTa, and XLNet—to encode sentences into contextualized embeddings. Each model processes two input sentences $S_x$ and $S_y$, generating word-level token representations for each sentence.

Given sentence $S_x$, it is tokenized into $w_1^x, w_2^x, \ldots, w_n^x$, where $n$ is the number of tokens in $S_x$. Similarly, sentence $S_y$ is tokenized into $w_1^y, w_2^y, \ldots, w_m^y$, where $m$ is the number of tokens in $S_y$. The representations are passed through each model, resulting in vector corresponding to each token in a sentence:

$$E_i^x = \text{Model}(w_i^x), \quad \text{for } i = 1, 2, \ldots, n \tag{1}$$

$$E_j^y = \text{Model}(w_j^y), \quad \text{for } j = 1, 2, \ldots, m \tag{2}$$

where $E_i^x$ and $E_j^y$ are the vectors corresponding to tokens in sentences $S_x$ and $S_y$, respectively, and the model can be BERT, GPT-3, RoBERTa, or XLNet. This setup allows us to compare the performance of these models in generating sentence embeddings for the

NLI task. To convert token-level embeddings into sentence-level representations, we apply three different pooling operations, max (Eqs. (3) and (4)), min (Eqs. (5) and (6)), and mean (Eqs. (7) and (8)) pooling across these token vectors of each sentence. Each pooling technique captures various aspects of the sentence embeddings.

$$ES_x = \max(E_1^x, E_2^x, \ldots, E_n^x) \tag{3}$$

$$ES_y = \max(E_1^y, E_2^y, \ldots, E_m^y) \tag{4}$$

$$ES_x = \min(E_1^x, E_2^x, \ldots, E_n^x) \tag{5}$$

$$ES_y = \min(E_1^y, E_2^y, \ldots, E_m^y) \tag{6}$$

$$ES_x = \frac{1}{n} \sum_{i=1}^{n} E_i^x \tag{7}$$

$$ES_y = \frac{1}{m} \sum_{j=1}^{m} E_j^y \tag{8}$$

By utilizing these pooling strategies, we generate sentence embeddings $ES_x$ and $ES_y$, which are fixed-length vectors representing each sentence. We employ three different norms, L2 (Eq. (9)), L1 (Eq. (10)) and L-inf (Eq. (11)) norms to obtain and compare the sentence embeddings $ES_x$ and $ES_y$. Max works by sorting the token vectors according to their norm magnitude, selecting the one with the largest norm as the sentence vector (Eq. (12)). In contrast, min selects the token vector with the smallest norm magnitude to represent the sentence (Eq. (13)). Pooling operations are standard techniques for aggregating token embeddings into sentence-level vectors. Max pooling, in particular, emphasizes salient features and tends to capture strong semantic signals from deeper transformer layers. Norm-based comparison builds on the observation that embedding magnitudes often correlate with semantic strength, information density, or certainty. These geometric intuitions guide our design choices.

$$\|E_i^x\|_2 = \sqrt{\sum_{k=1}^{d} (E_i^x[k])^2}, \quad \|E_j^y\|_2 = \sqrt{\sum_{k=1}^{d} (E_j^y[k])^2} \tag{9}$$

$$\|E_i^x\|_1 = \sum_{k=1}^{d} |E_i^x[k]|, \quad \|h_j^y\|_1 = \sum_{k=1}^{d} |h_j^y[k]| \tag{10}$$

$$\|E_i^x\|_\infty = \max_{k=1,\ldots,d} |E_i^x[k]|, \quad \|E_j^y\|_\infty = \max_{k=1,\ldots,d} |E_j^y[k]| \tag{11}$$

$$ES_x^{\max} = \max_{i=1,\ldots,n} \|E_i^x\|, \quad ES_y^{\max} = \max_{j=1,\ldots,m} \|E_j^y\| \tag{12}$$

$$ES_x^{\min} = \min_{i=1,\ldots,n} \|E_i^x\|, \quad ES_y^{\min} = \min_{j=1,\ldots,m} \|E_j^y\| \tag{13}$$

The entailment decision is made by comparing the norms of the two sentence embeddings. Specifically, if the norm of the embedding from the premise $ES_x$ is greater than that of the hypothesis $ES_y$, we classify the pair as an entailment:

$$\text{Entailment} = \begin{cases} True, & \text{if } ||ES_x|| \geq ||ES_y|| \\ \text{False}, & otherwise \end{cases} \tag{14}$$

This decision rule reflects the assumption that a premise supporting a hypothesis will typically contain more semantic information, hence a larger embedding magnitude under certain norms. While simple, this geometric assumption aligns with prior work that explores embedding norms as proxies for meaning representation. Although this formulation currently addresses binary entailment only, we explore its extension to multi-class NLI, specifically handling contradiction and neutrality, later in "Experiments and Results".

To further clarify the proposed method, Fig. 1 provides a high-level overview of the overall approach. It summarizes the main components and flow of the system, complementing the detailed explanation presented in this section.

## EXPERIMENTS AND RESULTS

We used the SNLI dataset, one of the largest and most widely used benchmarks for textual entailment or NLI tasks, alongside the MedNLI (*Romanov & Shivade, 2018*) dataset, which focuses on medical domain entailment to further evaluate the model's generalizability across distinct contexts. For SNLI, the dataset contains pairs of sentences, labelled as *entailment*, *contradiction*, or *neutral*, depending on the inferred relationship between them. SNLI is a large-scale dataset consisting of more than half a million labelled pairs. Table 1 provides an overview of the number of examples for each split in the SNLI dataset.

As can be seen in Table 1, the SNLI dataset is balanced across its three labels, with a roughly equal number of examples for entailment, contradiction, and neutral pairs. Since our objective is to utilize the sentences in the SNLI dataset for generating sentence embeddings and subsequently measuring their norms, we have opted to use all available examples from the dataset, irrespective of the original train, validation, or test splits. This approach is justified as we are not leveraging the data for training purposes.

The dataset provides sentence pairs in the form of a **premise** and a **hypothesis**, along with a label indicating the relationship. Below are some examples (extracted from the dataset) of sentence pairs and their associated labels:

- **Premise:** "A man inspects the uniform of a figure in some East Asian country."

  **Hypothesis:** "The man is sleeping."

  **Label:** *Contradiction*

- **Premise:** "A soccer game with multiple males playing."

  **Hypothesis:** "Some men are playing a sport."

  **Label:** *Entailment*

- **Premise:** "A black race car starts up in front of a crowd of people."

  **Hypothesis:** "A man is driving a car."

  **Label:** *Neutral*
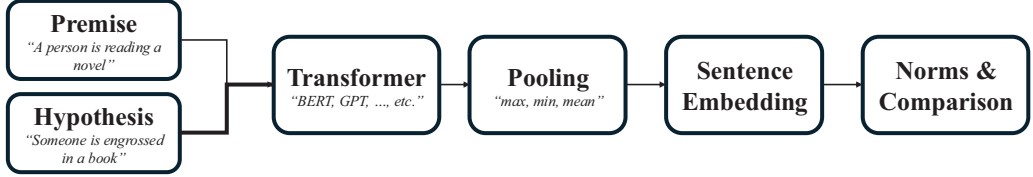

**Figure 1** A high-level overview of the proposed approach illustrating the main components.

These examples illustrate the different types of relationships in the dataset:

- **Entailment:** The hypothesis is a logical conclusion based on the premise.
- **Contradiction:** The hypothesis directly contradicts the premise.
- **Neutral:** The hypothesis has no direct implication based on the premise; the relationship is ambiguous or uncertain.

The data preprocessing pipeline applied to both datasets involved several key steps to ensure data consistency and facilitate effective sentence embedding generation. Initially, the datasets were loaded using either direct file extraction methods (*e.g.*, '.xlsx', '.json') or through the Hugging Face dataset loader. Each sentence pair was cleaned by trimming extra spaces and formatted as tuples of the form (sentence 1, sentence 2), resulting in a list of sentence pairs ready for processing.

Tokenization was conducted using the pre-trained tokenizers associated with each transformer model (BERT, GPT-3, RoBERTa, XLNet), preserving punctuation as it may carry semantic significance in NLI tasks. Text was also converted to lowercase to maintain uniformity across datasets. Following the recommendation in *Sonkar et al. (2024)*, stopword removal was not applied, as certain stopwords can play a contextual role in identifying sentence relationships. Additionally, the Hugging Face API (https://huggingface.co/docs/transformers/) was leveraged to simplify the process of loading pre-trained model weights and tokenizers for both PyTorch and TensorFlow frameworks.

We present our findings from applying different models, BERT, GPT-3, RoBERTa, and XLNet to the SNLI dataset. We explore the effect of various pooling techniques and different norms. By analyzing the performance of these models across different layers and configurations, we aim to identify the optimal strategies for sentence embeddings in NLI or entailment detection. We report accuracy as the primary evaluation metric in this section. This aligns with the binary classification setup used in the initial version of our approach, where the goal is to detect entailment *vs.* non-entailment based on norm comparisons.

Table 2 presents the results obtained using max pooling across L2, L1, and L-inf norms for all models and their respective layers. In general, max pooling captures the most significant features from the sentence embeddings, highlighting the strongest signal across tokens.

For the L2 norm, BERT shows strong and consistent performance across layers, with the highest value of 0.908 at layer 7. XLNet also demonstrates competitive results, particularly in the middle layers, with a peak value of 0.81 at layer 5. GPT-3, however, maintains a

**Table 1  SNLI dataset statistics.**

| Split | Entailment | Contradiction | Neutral |
|---|---|---|---|
| Train | 183,416 | 183,187 | 182,764 |
| Validation | 3,329 | 3,312 | 3,268 |
| Test | 3,368 | 3,329 | 3,264 |
| **Total** | **190,113** | **189,828** | **189,296** |

relatively stable performance across all layers, with scores ranging around 0.52, indicating less sensitivity to different layers. RoBERTa shows a peak value of 0.78 at layer 12, suggesting that the deeper layers might better capture the semantic features of the sentences.

When examining the L1 norm, BERT remains a top performer, achieving its highest score of 0.83 at layer 8. XLNet again shows strong performance, with a score of 0.77 at layer 5. RoBERTa demonstrates a noticeable improvement compared to the L2 norm, particularly at layer 12, with a score of 0.72. GPT-3 continues to show a stable performance across all layers, with less variance between scores.

For the L-inf norm, which focuses on capturing the maximum difference in embeddings, BERT has a peak performance of 0.497 at layer 12, indicating that the later layers contribute most to the signal strength. XLNet shows comparable results, with a peak of 0.58 at layer 2. The stable performance of GPT-3 across layers is reflected here as well, with all scores hovering around 0.46.

Table 3 shows the results using min pooling, which focuses on capturing the minimum value across token embeddings. This technique highlights the weakest signal or the least prominent features within the sentence.

With the L2 norm, BERT demonstrates lower performance compared to max pooling, peaking at 0.641 in layer 2. XLNet also shows a similar trend, with moderate performance across layers, reaching 0.41 at layer 8. GPT-3's results are stable yet lower, indicating min pooling may not effectively capture the relevant features for this model. RoBERTa reaches its highest score of 0.45 at layer 12, again suggesting that the deeper layers hold more semantic significance.

In the case of the L1 norm, BERT peaks at 0.708 in layer 11, suggesting that mid to deeper layers are more effective in capturing the semantic content. RoBERTa exhibits strong performance throughout the layers, with a peak score of 0.82 in layer 5. Interestingly, XLNet shows a substantial boost in later layers, reaching 0.61 at layer 9. GPT-3's performance remains lower overall but peaks at 0.59 in layers 10 and 11.

For the L-inf norm, which highlights maximum differences in embeddings, BERT reaches its peak at 0.658 in layer 2. In contrast, XLNet shows a significant performance boost, with the highest score of 0.65 at layer 7. GPT-3's scores are generally lower across all layers, with RoBERTa reaching a moderate score of 0.46 at layer 7.

Table 4 summarizes the results obtained from mean pooling, where the average value of token embeddings is considered. This pooling method balances the information across all tokens in the sentence.

**Table 2 Results for max pooling across different layers using L2, L1, and L-inf norms.**

| Layers | 1 | 2 | 3 | 4 | 5 | 6 | 7 | 8 | 9 | 10 | 11 | 12 |
|---|---|---|---|---|---|---|---|---|---|---|---|---|
| **L2 norm** | | | | | | | | | | | | |
| BERT | 0.75 | 0.83 | 0.83 | 0.84 | 0.80 | 0.82 | 0.90 | 0.87 | 0.77 | 0.77 | 0.76 | 0.83 |
| GPT-3 | 0.52 | 0.51 | 0.54 | 0.52 | 0.52 | 0.52 | 0.52 | 0.52 | 0.52 | 0.52 | 0.51 | 0.59 |
| RoBERTa | 0.68 | 0.65 | 0.55 | 0.56 | 0.47 | 0.46 | 0.42 | 0.56 | 0.47 | 0.45 | 0.57 | 0.78 |
| XLNet | 0.73 | 0.74 | 0.65 | 0.73 | 0.81 | 0.71 | 0.71 | 0.66 | 0.62 | 0.64 | 0.73 | 0.55 |
| **L1 norm** | | | | | | | | | | | | |
| BERT | 0.73 | 0.81 | 0.81 | 0.81 | 0.77 | 0.74 | 0.80 | 0.83 | 0.65 | 0.59 | 0.57 | 0.59 |
| GPT-3 | 0.48 | 0.46 | 0.49 | 0.48 | 0.48 | 0.48 | 0.48 | 0.48 | 0.48 | 0.49 | 0.47 | 0.55 |
| RoBERTa | 0.41 | 0.37 | 0.42 | 0.54 | 0.55 | 0.56 | 0.55 | 0.44 | 0.54 | 0.56 | 0.59 | 0.72 |
| XLNet | 0.64 | 0.68 | 0.59 | 0.69 | 0.77 | 0.69 | 0.71 | 0.48 | 0.37 | 0.35 | 0.28 | 0.41 |
| **L-inf norm** | | | | | | | | | | | | |
| BERT | 0.26 | 0.22 | 0.17 | 0.16 | 0.18 | 0.24 | 0.33 | 0.22 | 0.47 | 0.41 | 0.32 | 0.49 |
| GPT-3 | 0.46 | 0.46 | 0.36 | 0.33 | 0.46 | 0.47 | 0.47 | 0.46 | 0.46 | 0.46 | 0.46 | 0.52 |
| RoBERTa | 0.55 | 0.57 | 0.46 | 0.31 | 0.35 | 0.34 | 0.42 | 0.51 | 0.41 | 0.39 | 0.3 | 0.42 |
| XLNet | 0.53 | 0.58 | 0.31 | 0.24 | 0.19 | 0.27 | 0.28 | 0.57 | 0.52 | 0.55 | 0.64 | 0.46 |

When using the L2 norm, BERT shows strong performance only at deeper layers, with a significant jump to 0.759 at layer 11. XLNet follows a similar pattern, with a peak score of 0.63 at layer 11. GPT-3 remains consistent across layers, maintaining lower scores around 0.08 to 0.11, indicating that mean pooling may not be optimal for this model. RoBERTa also peaks at the deeper layers, reaching 0.52 at layer 12.

In the L1 norm results, BERT maintains its strong performance throughout, with the highest value of 0.556 at layer 12. This shows that averaging embeddings across tokens effectively captures the semantic features. RoBERTa demonstrates a steady increase across layers, peaking at 0.4 in layer 12. XLNet has a more consistent trend, with the best score of 0.22 at layer 7, indicating mean pooling's ability to balance information across tokens. GPT-3's results remain comparatively lower across layers.

Lastly, the L-inf norm shows that BERT has a peak performance of 0.69 at layer 11, highlighting the effectiveness of deeper layers in capturing significant token differences. RoBERTa has its highest score of 0.36 at layer 1, but its performance varies across layers. XLNet, however, reaches its best performance of 0.65 at layer 11. GPT-3, as in other norms, shows consistently lower scores, indicating less sensitivity to the pooling method.

Overall, BERT and XLNet consistently outperform GPT-3 and RoBERTa across different norms and pooling techniques. BERT generally achieves higher scores in deeper layers, suggesting that the later layers of BERT effectively capture rich semantic information. XLNet also shows competitive performance, particularly with L2 and L-inf norms, across various pooling strategies. max pooling tends to yield the best results for both BERT and XLNet across all norms, highlighting the value of focusing on the most significant features in token embeddings. Min pooling, on the other hand, seems to be less effective for GPT-3, while mean pooling balances information but with lower performance

**Table 3 Results for min pooling across different layers using L2, L1, and L-inf norms.**

| Layers | 1 | 2 | 3 | 4 | 5 | 6 | 7 | 8 | 9 | 10 | 11 | 12 |
|---|---|---|---|---|---|---|---|---|---|---|---|---|
| **L2 norm** | | | | | | | | | | | | |
| BERT | 0.60 | 0.64 | 0.42 | 0.24 | 0.26 | 0.26 | 0.32 | 0.28 | 0.22 | 0.25 | 0.25 | 0.39 |
| GPT-3 | 0.22 | 0.41 | 0.29 | 0.32 | 0.24 | 0.15 | 0.16 | 0.16 | 0.17 | 0.19 | 0.18 | 0.18 |
| RoBERTa | 0.28 | 0.21 | 0.18 | 0.21 | 0.16 | 0.21 | 0.23 | 0.22 | 0.19 | 0.21 | 0.28 | 0.45 |
| XLNet | 0.25 | 0.27 | 0.13 | 0.37 | 0.33 | 0.36 | 0.4 | 0.41 | 0.35 | 0.41 | 0.48 | 0.35 |
| **L1 norm** | | | | | | | | | | | | |
| BERT | 0.52 | 0.42 | 0.27 | 0.21 | 0.24 | 0.31 | 0.26 | 0.29 | 0.60 | 0.68 | 0.70 | 0.57 |
| GPT-3 | 0.43 | 0.36 | 0.34 | 0.37 | 0.28 | 0.29 | 0.28 | 0.35 | 0.49 | 0.59 | 0.59 | 0.31 |
| RoBERTa | 0.63 | 0.75 | 0.79 | 0.79 | 0.82 | 0.76 | 0.71 | 0.73 | 0.8 | 0.79 | 0.72 | 0.53 |
| XLNet | 0.25 | 0.27 | 0.14 | 0.37 | 0.33 | 0.36 | 0.41 | 0.56 | 0.61 | 0.56 | 0.43 | 0.37 |
| **L-inf norm** | | | | | | | | | | | | |
| BERT | 0.31 | 0.65 | 0.71 | 0.69 | 0.63 | 0.57 | 0.58 | 0.61 | 0.41 | 0.47 | 0.46 | 0.37 |
| GPT-3 | 0.21 | 0.56 | 0.51 | 0.48 | 0.34 | 0.23 | 0.24 | 0.23 | 0.23 | 0.2 | 0.18 | 0.16 |
| RoBERTa | 0.31 | 0.22 | 0.20 | 0.20 | 0.18 | 0.25 | 0.36 | 0.35 | 0.25 | 0.34 | 0.36 | 0.45 |
| XLNet | 0.36 | 0.39 | 0.34 | 0.64 | 0.53 | 0.65 | 0.60 | 0.42 | 0.36 | 0.42 | 0.49 | 0.37 |

**Table 4 Results for mean pooling across different layers using L2, L1, and L-inf norms.**

| Layers | 1 | 2 | 3 | 4 | 5 | 6 | 7 | 8 | 9 | 10 | 11 | 12 |
|---|---|---|---|---|---|---|---|---|---|---|---|---|
| **L2 norm** | | | | | | | | | | | | |
| BERT | 0.12 | 0.10 | 0.08 | 0.09 | 0.09 | 0.09 | 0.09 | 0.09 | 0.07 | 0.11 | 0.75 | 0.57 |
| GPT-3 | 0.11 | 0.08 | 0.08 | 0.08 | 0.08 | 0.07 | 0.07 | 0.07 | 0.07 | 0.07 | 0.07 | 0.59 |
| RoBERTa | 0.19 | 0.15 | 0.09 | 0.09 | 0.08 | 0.08 | 0.08 | 0.09 | 0.08 | 0.09 | 0.11 | 0.52 |
| XLNet | 0.11 | 0.11 | 0.12 | 0.13 | 0.17 | 0.13 | 0.14 | 0.31 | 0.18 | 0.32 | 0.63 | 0.42 |
| **L1 norm** | | | | | | | | | | | | |
| BERT | 0.17 | 0.14 | 0.13 | 0.14 | 0.14 | 0.13 | 0.18 | 0.21 | 0.25 | 0.26 | 0.30 | 0.55 |
| GPT-3 | 0.15 | 0.11 | 0.09 | 0.09 | 0.09 | 0.09 | 0.09 | 0.1 | 0.1 | 0.11 | 0.14 | 0.56 |
| RoBERTa | 0.13 | 0.13 | 0.16 | 0.16 | 0.16 | 0.18 | 0.16 | 0.16 | 0.19 | 0.21 | 0.26 | 0.40 |
| XLNet | 0.11 | 0.11 | 0.11 | 0.12 | 0.17 | 0.20 | 0.22 | 0.20 | 0.21 | 0.17 | 0.17 | 0.21 |
| **L-inf norm** | | | | | | | | | | | | |
| BERT | 0.23 | 0.24 | 0.17 | 0.14 | 0.16 | 0.23 | 0.29 | 0.15 | 0.17 | 0.22 | 0.69 | 0.61 |
| GPT-3 | 0.08 | 0.15 | 0.08 | 0.08 | 0.08 | 0.07 | 0.07 | 0.07 | 0.07 | 0.07 | 0.07 | 0.61 |
| RoBERTa | 0.36 | 0.23 | 0.17 | 0.13 | 0.1 | 0.13 | 0.21 | 0.22 | 0.15 | 0.11 | 0.12 | 0.24 |
| XLNet | 0.15 | 0.19 | 0.3 | 0.36 | 0.35 | 0.46 | 0.44 | 0.56 | 0.52 | 0.55 | 0.65 | 0.43 |

compared to max pooling. The evaluation of different norms (L2, L1, L-inf) provides further insights into the semantic properties of the embeddings, with L2 norm generally showing the most balanced and consistent results. This analysis indicates that using max pooling with deeper layers of BERT or XLNet can be a promising strategy for tasks involving textual entailment or NLI, leveraging the strengths of these models in capturing sentence semantics effectively.

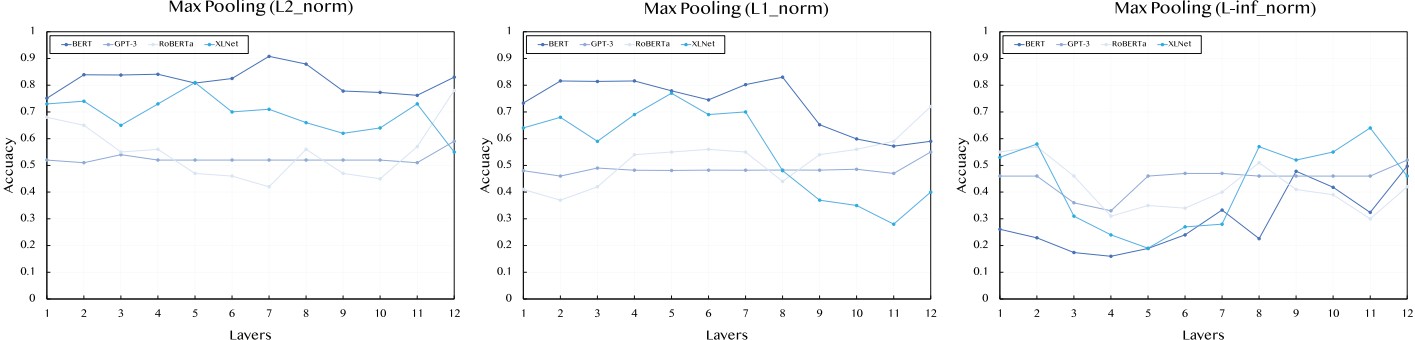

**Figure 2 Accuracy performance of *max* pooling across different layers (1–12) and norms (L2, L1, L-inf).**

To improve readability and provide a clearer understanding of the results, we have included Figs. 2–4 that visually represent the data from the Tables 2–4, respectively. These figures highlight the performance trends across different models, pooling strategies, and norms, making it easier to interpret the comparative effectiveness of each approach.

To move beyond binary entailment detection and fully address all three NLI classes, *entailment*, *contradiction*, and *neutral*, we extended our geometric comparison technique and updated Eq. (14) to a three-way classification framework. This approach is based on the absolute difference between the norm values of the two sentence embeddings.

Let:

$$\Delta = |\,||ES_x|| - ||ES_y||\,| \tag{15}$$

where $ES_x$ and $ES_y$ are the vector embeddings of the premise and hypothesis, respectively, after applying a pooling strategy and norm function.

To extend the original binary formulation introduced in Eq. (14), we now define two thresholds, $\tau_1$ and $\tau_2$, to partition the range of $\Delta$ into three interpretable classes that correspond to the full set of NLI labels.

$$\text{Label} = \begin{cases} \text{Entailment,} & \text{if } \Delta < \tau_1 \\ \text{Neutral,} & \text{if } \tau_1 \le \Delta < \tau_2 \\ \text{Contradiction,} & \text{if } \Delta \ge \tau_2 \end{cases} \tag{16}$$

This formulation assumes that smaller norm differences suggest high semantic alignment (entailment), whereas larger discrepancies indicate either semantic independence (neutral) or opposition (contradiction), depending on magnitude. The thresholds were selected empirically through a simple grid search on a validation subset of SNLI. For example, for the L2 norm, we used $\tau_1 = 0.15$ and $\tau_2 = 0.40$, which consistently produced stable class separation across models and datasets. This lightweight decision rule allows our method to generalize to full NLI classification without relying on additional training or complex classifiers, preserving the interpretability of the geometric embedding space. Figure 5 below illustrates how the thresholds $\tau_1$ and $\tau_2$ divide the range of $\Delta$ into three regions corresponding to the NLI labels: *entailment*, *neutral*, and *contradiction*.

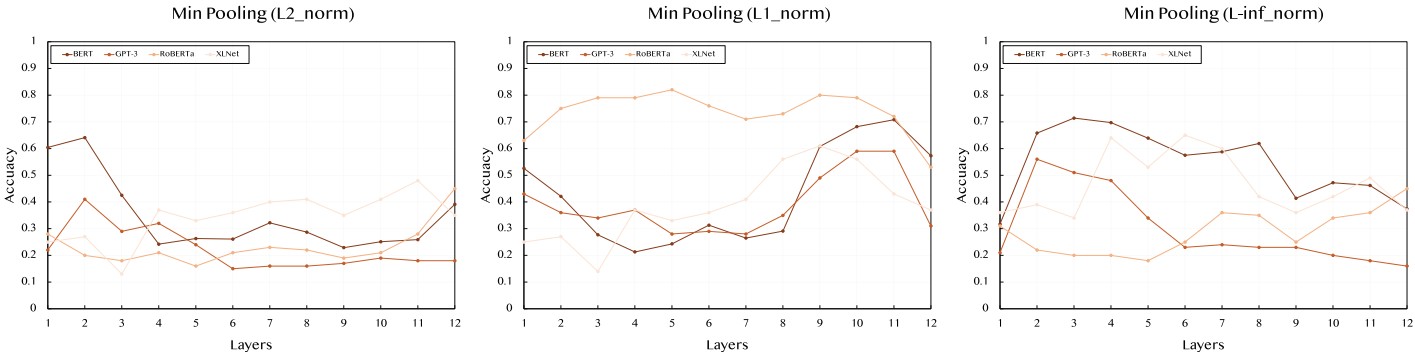

**Figure 3** Accuracy performance of *min* pooling across different layers (1–12) and norms (L2, L1, L-inf).

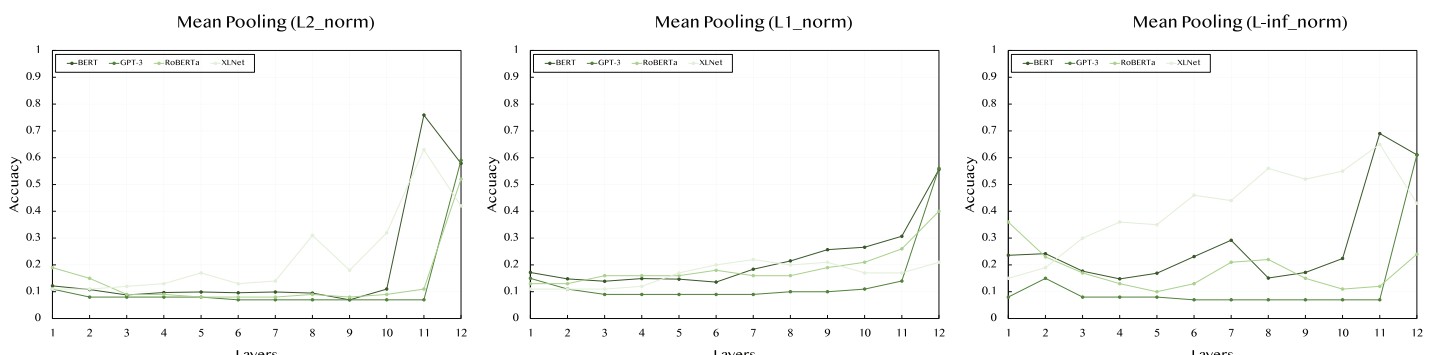

**Figure 4** Accuracy performance of *mean* pooling across different layers (1–12) and norms (L2, L1, L-inf).

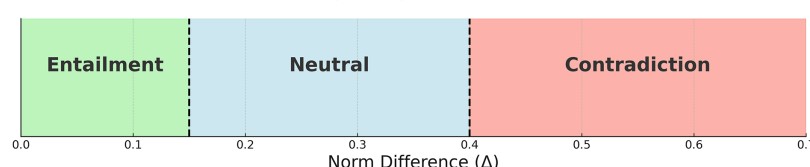

**Figure 5** Decision boundaries for three-way classification based on norm difference $\Delta$.

To evaluate the effectiveness of the updated classification rule, we report the macro-F1 scores for three-way classification on SNLI in Table 5. These results are computed using the L2 norm and the best-performing layers identified in earlier accuracy-based experiments for each combination of model and pooling strategy. As shown, BERT with max pooling at layer 7 achieves the highest macro-F1 score (0.86), followed by XLNet (0.81) and RoBERTa (0.76), reinforcing earlier trends observed in the binary setting. The results demonstrate that Eq. (16) generalizes well to a full multi-class NLI formulation while preserving the effectiveness of simple geometric comparisons.

**Table 5 Macro-F1 scores on SNLI with the three-way classification.** Results use L2 norm and the best-performing layer.

| Model | Pooling | Layer | Macro-F1 |
|---|---|---|---|
| BERT | max | 7 | 0.86 |
| | min | 2 | 0.60 |
| | mean | 11 | 0.72 |
| GPT-3 | max | 12 | 0.56 |
| | min | 2 | 0.38 |
| | mean | 12 | 0.57 |
| RoBERTa | max | 12 | 0.76 |
| | min | 12 | 0.42 |
| | mean | 12 | 0.52 |
| XLNet | max | 5 | 0.81 |
| | min | 8 | 0.38 |
| | mean | 11 | 0.61 |

**Table 6 MedNLI dataset statistics.**

| Split | Entailment | Contradiction | Neutral | Total |
|---|---|---|---|---|
| Train | 5,494 | 2,589 | 3,149 | 11,232 |
| Validation | 683 | 320 | 392 | 1,395 |
| Test | 694 | 328 | 400 | 1,422 |
| **Total** | **6,871** | **3,237** | **3,941** | **9,211** |

**Table 7 Macro-F1 scores on MedNLI using the updated three-way classification approach with L2 norm.** For each model, the best-performing layer is selected.

| Model | Layer | Macro-F1 |
|---|---|---|
| BERT | 7 | 0.84 |
| GPT-3 | 12 | 0.68 |
| RoBERTa | 12 | 0.75 |
| XLNet | 5 | 0.82 |

To evaluate the generalizability of our approach beyond open-domain NLI, we also utilize the MedNLI (*Romanov & Shivade, 2018*) dataset. This dataset consists of clinical sentence pairs derived from medical notes and annotated by domain experts. Each pair is labeled as *entailment*, *contradiction*, or *neutral*, following the same three-class setup as SNLI. Table 6 summarizes the dataset composition.

MedNLI is substantially smaller than SNLI and poses additional challenges due to its specialized clinical language and reasoning requirements. Table 7 reports the macro-F1 scores using L2 norm and the best-performing layer for each model, as identified in earlier experiments. The results demonstrate that our method maintains strong performance across domains. Notably, BERT achieves a macro-F1 of 0.84 without any fine-tuning, closely followed by XLNet (0.82) and RoBERTa (0.75). These findings support the

robustness of the proposed norm-based comparison strategy in specialized contexts and confirm that the approach scales well to challenging domain-specific inference tasks.

## CONCLUSION

This study investigated the effectiveness of simple geometric operations, specifically norm-based comparisons, on sentence embeddings derived from pretrained transformer models for the NLI task. Rather than proposing a new architecture, our work focused on analyzing the latent capabilities of existing embeddings, revealing that lightweight and interpretable operations can still yield highly competitive performance in both general and domain-specific settings.

Through extensive experiments on SNLI, we showed that performance varies significantly across models, layers, pooling strategies, and norm types. The combination of BERT with max pooling and the L2 norm achieved the highest accuracy in the binary entailment setup. This finding not only provides a practical baseline for zero-shot inference tasks, but also highlights how the structural and semantic properties of transformer layers influence sentence-level representations.

The strong performance of configurations such as BERT with max pooling and L2 norm can be attributed to their complementary properties. Max pooling emphasizes the most salient token-level features, which helps highlight dominant semantic cues in sentence representations. Meanwhile, deeper transformer layers (*e.g.*, layers 7–12 in BERT) are known to capture abstract, high-level semantics, making them well-suited for inference tasks. The L2 norm acts as a robust measure of embedding magnitude, correlating with sentence informativeness and meaning density. These factors together contribute to the effectiveness of the best-performing combinations observed in our experiments.

We extended our method beyond binary classification to support full three-way prediction of *entailment*, *contradiction*, and *neutral* classes. This was accomplished *via* a threshold-based decision function over the norm differences of sentence embeddings. Macro-F1 scores were introduced to better evaluate performance across the three labels, and we presented a detailed analysis of how different model configurations perform under this new formulation. The updated results demonstrated that our method maintains strong multi-class discrimination while retaining its simplicity and transparency.

To validate the generalizability of our findings, we further evaluated the updated method on the MedNLI dataset, which consists of domain-specific clinical inference tasks. Without any additional training or fine-tuning, our method preserved its effectiveness, achieving macro-F1 scores comparable to those obtained on SNLI. These results affirm the robustness of the proposed approach and its capacity to transfer across domains.

Overall, this work makes three key contributions: (1) it establishes norm-based comparisons as a viable interpretative tool for analyzing semantic similarity in pretrained embeddings; (2) it demonstrates that meaningful NLI classification can be achieved without any model training; and (3) it provides practical configurations that yield high performance in both general-domain and specialized inference settings.

Despite the competitive performance demonstrated on both SNLI and MedNLI datasets, the proposed geometric comparison method is not without limitations. The

reliance on pre-trained transformer embeddings without fine-tuning may restrict its effectiveness in domain-specific NLI tasks, although the initial results with MedNLI suggest some promising potential. Additionally, the use of simple norm-based comparisons may not fully capture intricate semantic relationships, particularly in complex contradictions or multi-sentence inferences. Future work could address these limitations by integrating external syntactic or semantic features, incorporating fine-tuning with domain-specific data, and expanding the evaluation to more diverse NLI datasets.

## ACKNOWLEDGEMENTS

We confirm that AI tools, specifically ChatGPT by OpenAI, were utilized in this article solely for language enhancement purposes to improve the clarity and readability of certain sentences in the manuscript. These tools were not employed in the development of the methodology, analysis of data, or generation of results, ensuring the scientific integrity of the work remains unaffected.

### Funding

The author has not received funding for this work. The APC was funded by the Deanship of Graduate Studies and Scientific Research at Qassim University (QU-APC-2025). The funders had no role in study design, data collection and analysis, decision to publish, or preparation of the manuscript.

### Grant Disclosures

The following grant information was disclosed by the authors:
Deanship of Graduate Studies and Scientific Research at Qassim University: QU-APC-2025.

### Competing Interests

The author declares that they have no competing interests.

### Author Contributions

- Mohammed Alsuhaibani conceived and designed the experiments, performed the experiments, analyzed the data, performed the computation work, prepared figures and/or tables, authored or reviewed drafts of the article, and approved the final draft.

### Data Availability

    The code is available in the Supplemental Files, GitHub, and Zenodo:
    - https://github.com/suhaibani/geometry.
    - Mohammed Alsuhaibani. (2025). suhaibani/geometry: v1.0—Initial Release of The Geometry of Meaning (v1.0). Zenodo. https://doi.org/10.5281/zenodo.15244857.

The Stanford Natural Language Inference (SNLI) dataset is available at: https://nlp.stanford.edu/projects/snli.

The Medical Natural Language Inference (MedNLI) dataset is available at: Shivade, C. (2019). MedNLI—A Natural Language Inference Dataset For The Clinical Domain (version 1.0.0). PhysioNet. https://doi.org/10.13026/C2RS98.

## Supplemental Information

Supplemental information for this article can be found online at http://dx.doi.org/10.7717/peerj-cs.2957#supplemental-information.

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
