# Peer review of "The geometry of meaning: evaluating sentence embeddings from diverse transformer-based models for natural language inference"

_PeerJ Computer Science, doi:10.7717/peerj-cs.2957_

## Round 0.1 · original submission · Major Revisions

The primary concerns include the lack of theoretical justification for using norms to determine entailment, missing comparisons with prior work, and reliance on a single dataset (SNLI), limiting generalizability. Additionally, the explanation of equation 14 and the evaluation metrics require clarification. Formatting inconsistencies in figures, tables, and references should be addressed, and redundant data presentation should be avoided. Expanding the discussion on why certain configurations perform better and incorporating a baseline comparison would strengthen the study. Lastly, if the method only applies to entailment cases, the scope should be broadened to include contradiction and neutral labels or explicitly justified.

Reviewer 1 ·

Basic reporting

The paper conducts comprehensive experiments with different norms and pooling techniques to evaluate the word embeddings for the task of natural language inference. However, there are several shortcomings in the study:

- In the introduction section, the authors discussed NLI but the references are incorrect, for readability “text summarization Laban et al. (2022)” should be in parenthesis “text summarization (Laban et al., 2022)”. Similarly, the authors' names are not part of the sentences in many places in the paper but appear without parenthesis.
- In the related work section, the present tense should be avoided, unless the author presents his/her point of view.
- The proposed approach should offer a diagram for clarity about the process. Currently, it lacks clarity.
- The information in tables and figures is duplicated. This should be avoided.
- The figures take up a lot of page space. Such figures should be displayed in two columns for the economy.

Experimental design

- The statement "The entailment is then determined by comparing the norms of the two sentence embeddings" and equation 14 for determining entailment are unclear. How comparison of the two sentences' embeddings determine if the label is entailment or contradiction?

Validity of the findings

- NLI is a well-known task and the SNLI dataset has been used in several studies. However, the manuscript does not compare the performance of the task with previous studies.
- The evaluation metrics of the proposed methodology are not clear.

Reviewer 2 ·

Basic reporting

The study explores an important area: evaluating sentence embeddings for Natural Language Inference (NLI) using transformer models.

Following are my comments

-Limited to a single dataset (SNLI), which can not generalize to other NLI tasks or domains. Exploring datasets like MultiNLI or other multilingual NLI datasets could provide a broader scope.
-The geometric approach to comparing embeddings (via norms) is interesting but lacks theoretical grounding or evidence of applicability beyond the NLI context.
-Abstract: The phrase "reaching %90" is an error; it should be "reaching 90%."
-Introduction: The sentence "This process requires an understanding of language syntax and semantics" could be expanded for clarity on the type of understanding needed.
-Figures and Tables: Titles and captions are inconsistent and need better formatting for clarity (e.g., "Performance of max pooling" lacks detail about norms or layer depth

Experimental design

-The study does not address the limitations of the proposed approach sufficiently. For example, it lacks a discussion on potential overfitting or biases from relying solely on SNLI.
-The conclusions are valid but simplistic. There is no exploration of why certain configurations work better or how these results compare with state-of-the-art NLI models.
A more detailed justification for the choice of pooling strategies or norms is missing. For example, why are these norms considered effective for NLI tasks?

Validity of the findings

Justify the use of specific norms and pooling strategies, and explore theoretical implications.

Reviewer 3 ·

Basic reporting

This paper investigates how different transformer-based models generate sentence embeddings for the Natural Language Inference (NLI) task. The authors conducted a comprehensive evaluation of the NLI accuracy regarding different settings of the pooling and norm techniques in multiple layers. While the study offers a set of straightforward evaluations, the contributions appear to be somewhat incremental, focusing on variations of current simple parameter settings on a single task and dataset rather than introducing fundamentally technical solutions.

Experimental design

1. The study conducts rigorous ablation tests to explore multiple transformer-based models, with pooling strategies (max, min, mean) and norm-based measures (L1, L2, L-inf), offering a broad comparative analysis. The methods are described with sufficient details to replicate. Still, the results (e.g., best model's layer that achieves the highest accuracy in NLI task) seem random without intuitive explanation and further causal analysis.

2. The findings highlight that max pooling with the L2 norm, particularly in deeper layers of the BERT model, yields the best performance for NLI, which could be helpful in constructing better optimizing strategies for sentence embedding techniques. However, according to equation 14, it seems this embedding-based mechanism only applies to the entailment but not contradiction and neutral scenarios.

3. Though data instances from the SNLI dataset already possess good quality, authors should involve necessary data pre-processing techniques to best suit the needs of their proposed embedding-based classification. For example, removing the stopwords can be a possible option for pre-processing. Necessary declaration and details should be involved.

Validity of the findings

1. Including a GitHub repository with code and dataset references enhances reproducibility and enables easy replication of the experiments.

2. If the experiments are only conducted to distinguish the entailment cases, the proposed technique would be less insightful; if the experiments are conducted for all three NLI tasks, then adopting f1-score is a better choice than accuracy measurement.

3. With recent advancements in language models and AI-based systems, the proposed technique lacks a valid baseline against which to compare its performance and cost-efficiency. For example, a fine-tuned small language model like GPT-2-small can effectively conduct the same classification tasks.

4. The limitation section mentions the point of using only one dataset, which should be resolved but not simply mentioned in the limitation part.

Additional comments

It would be good to have a baseline approach to compare. If you just test the entailment scenario, it is better to include contradiction and neutral scenarios of the NLI task.

---

## Round 0.2 · Minor Revisions

Please address all remaining points, especially regarding details for improving the replicability of the work and discussion of limitations.

Reviewer 1 ·

Basic reporting

The manuscript is well-structured, and the authors have addressed reviewers' comments. However, the authors should discuss the results on both datasets in the abstract, including F1 scores.

Experimental design

NA

Validity of the findings

NA

Additional comments

The authors should clearly describe the limitations and future work of the study.

Reviewer 3 ·

Basic reporting

This paper investigates how different transformer-based models generate sentence embeddings for the Natural Language Inference (NLI) task. The authors conducted a comprehensive evaluation of the NLI accuracy regarding different settings of the pooling and norm techniques in multiple layers.

Generally, authors addressed most of my previous concerns after the major revisions.

Experimental design

1. The study conducts rigorous ablation tests to explore multiple transformer-based models, with pooling strategies (max, min, mean) and norm-based measures (L1, L2, L-inf), offering a broad comparative analysis. The methods are described with sufficient detail to replicate.

- In the revised version, the authors included an extra dataset and further analysis to make the results more interpretable.

2. The findings highlight that max pooling with the L2 norm, particularly in deeper layers of the BERT model, yields the best performance for NLI, which could be helpful in constructing better optimizing strategies for sentence embedding techniques.

- In the revised version, the authors address my previous concern by including the Neutral and Contradiction scenarios in the NLI tasks.

3. Though data instances from the SNLI dataset already possess good quality, authors should involve necessary data pre-processing techniques to best suit the needs of their proposed embedding-based classification. For example, removing the stopwords can be a possible option for pre-processing. Necessary declarations and details should be included.

- However, this part is not disclosed in the revised version.

Validity of the findings

1. Including a GitHub repository with code and dataset references enhances reproducibility and enables easy replication of the experiments.

- In the revised version, this has been addressed.

2. If the experiments are only conducted to distinguish the entailment cases, the proposed technique would be less insightful; if the experiments are conducted for all three NLI tasks, then adopting the F1-score is better than the accuracy measurement.

- In the revised version, this has been addressed.

3. With recent advancements in language models and AI-based systems, the proposed technique lacks a valid baseline against which to compare its performance and cost-efficiency. For example, a fine-tuned small language model like GPT-2-small can effectively conduct the same classification tasks.

- More models are included in the revised version.

4. The limitation section mentions the point of using only one dataset, which should be resolved, but not simply mentioned in the limitation section.

- In the revised version, this has been addressed.

Additional comments

Still, it would be better to disclose further details (code, processed datasets, etc.) to improve the replicability of this work.

---

## Round 0.3 · accepted · Accept

The authors have addressed all comments and ready for publication. The newly added text contains some typos like wrong quote characters so I'd urge the authors to carefully proofread the manuscript again.